# Probe-Based Confocal Laser Endomicroscopy versus White-Light Endoscopy with Narrow-Band Imaging for Predicting and Collecting Residual Cancer Tissue in Patients with Gastric Cancer Receiving Chemotherapy

**DOI:** 10.3390/cancers14174319

**Published:** 2022-09-03

**Authors:** Yuna Kim, Hyunki Kim, Minkyu Jung, Sun Young Rha, Hyun Cheol Chung, Sang Kil Lee

**Affiliations:** 1Division of Gastroenterology, Department of Internal Medicine, Severance Hospital, Yonsei University College of Medicine, 50-1 Yonsei-ro, Seodaemun-gu, Seoul 03722, Korea; 2Department of Pathology, Severance Hospital, Yonsei University College of Medicine, Seoul 03722, Korea; 3Division of Medical Oncology, Department of Internal Medicine, Severance Hospital, Yonsei University College of Medicine, Seoul 03722, Korea

**Keywords:** probe-based confocal laser endomicroscopy, white-light endoscopy, biopsy, gastric cancer, chemotherapy

## Abstract

**Simple Summary:**

Collecting appropriate gastric cancer (GC) tissues is critical for molecular biology research or the development of new target drugs for cases resistant to chemotherapy. Chemotherapy may reduce or alter the distribution of GC tissue on the surface, making the detection of GC tissue during upper endoscopy challenging. Our study showed that probe-based confocal laser endomicroscopy (pCLE) is superior to white-light endoscopy (WLE) with magnifying narrow-band imaging (M-NBI) in terms of accuracy for diagnosing residual cancer in GC patients receiving chemotherapy. pCLE might be considered when it is necessary to confirm the presence of residual cancer and get tissue samples from GC patients receiving chemotherapy.

**Abstract:**

In cases of progression despite chemotherapy, collecting gastric cancer (GC) tissues might be helpful for molecular biology research or the development of new target drugs for treating cases that are refractory to chemotherapy. Chemotherapy, however, may reduce or alter the distribution of GC tissue on the surface, making the detection of GC tissue during upper endoscopy challenging. Probe-based confocal laser endomicroscopy (pCLE) is a new technology that enables histological diagnosis by magnifying the mucous membrane to a microscopic level. Here, we evaluated whether pCLE could increase the yield of endoscopic biopsy for GC compared to white-light endoscopy (WLE) with magnifying narrow-band imaging (M-NBI) in GC patients receiving chemotherapy with its powerful imaging technique. Patients underwent WLE/M-NBI and pCLE for the detection of residual GC for the purpose of response evaluation or clinical trial registration. After WLE/M-NBI and pCLE, each residual GC lesion was biopsied for histological analysis. A total of 23 patients were enrolled between January 2018 and June 2020. Overall, pCLE showed significantly higher sensitivity and negative predictive value than WLE/M-NBI. The accuracy of pCLE was superior to that of WLE/M-NBI. Moreover, pCLE showed better predictive ability for residual GC than WLE/M-NBI, while WLE/M-NBI and pCLE showed inconsistent results. pCLE diagnosed residual GC more accurately than WLE/M-NBI, which resulted in an increased number of GC tissues collected during the endoscopic biopsy.

## 1. Introduction

As the sixth most common cancer and fifth leading cause of cancer-related deaths worldwide, gastric cancer (GC) causes a considerable health burden [1,2]. As GC is biologically and genetically heterogeneous, with complex carcinogenesis at the molecular level, many investigators have attempted to classify heterogeneous populations of GC patients into homogeneous subgroups [3,4]. For this purpose, GC tissues must be obtained through surgery or endoscopic biopsy. In particular, it is essential to obtain specimens via endoscopic biopsy in inoperable GC patients. Genetic profiling of GC is mandatory before initiating chemotherapy. In the case of progression despite chemotherapy, collecting proper GC tissues is essential for molecular biology research or the development of new target drugs for cases that are refractory to chemotherapy.

To enable the targeted therapy of GC based on its molecular characteristics, high-yield biopsies of cancer tissue are mandatory. Owing to the development of optical and electronic technology, diagnostic techniques such as conventional white-light endoscopy (WLE) and virtual chromoendoscopy, including magnifying narrow-band imaging (M-NBI), are evolving; however, the diagnostic yield of a single endoscopic biopsy using WLE is suboptimal [5]. WLE/M-NBI has been reported to have a sensitivity of 93% to 97% for distinguishing GC from benign lesions, while its specificity is 84% to 97% [6,7].

In particular, for research on resistance to chemotherapy or new therapeutic agents, gastric tissue samples must be collected from patients who previously received chemotherapy; however, it is challenging to accurately localize GC tissues using endoscopy alone [8]. Surface GC tissues are replaced by ulcerated and hyperplastic tissues in response to chemotherapy, which reduces the distribution of GC tissue on the surface—thereby reducing the possibility of collecting GC tissue via endoscopic biopsy. Furthermore, it is highly likely that the distribution of the initial GC tissues decreases or changes in response to systemic chemotherapy.

Probe-based confocal laser endomicroscopy (pCLE) is a powerful imaging technique that allows the acquisition of high-resolution in vivo images at the cellular and microvascular levels [9]. The main clinical benefits of pCLE include enabling a real-time histopathological diagnosis, aiding decision-making, and superior diagnostic accuracy [10]. In fact, pCLE has high sensitivity and specificity compared to traditional WLE for gastrointestinal cancers, including esophageal cancer, GC, and colorectal cancer [9,11,12]. It can also be used to guide endoscopic biopsies of a target lesion, reduce the number of biopsies required for histological confirmation, and increase the content and purity of cancer tissue in biopsy specimens [3,13]. In our previous study, pCLE-targeted biopsies resulted in a significantly higher content of cancer cells in biopsy specimens than biopsy via conventional endoscopy. In particular, pCLE showed definitive superiority in cases of GC with undifferentiated histology, which are expected to have a low content of intraepithelial cancer cells [5].

Therefore, this prospective study aimed to determine whether adding pCLE to WLE/M-NBI endoscopy could improve the prediction and collection of residual cancer tissues in GC patients receiving chemotherapy.

## 2. Materials and Methods

### 2.1. Patients

This prospective single-arm study enrolled 55 patients but prematurely ended owing to a low recruitment rate. Between February 2018 and May 2021, 28 patients aged 20–80 years who underwent chemotherapy for advanced GC were enrolled. The included patients underwent endoscopy and additional biopsy to evaluate their response to chemotherapy, after which they could choose a new chemotherapy regimen or register for the clinical study. Five patients dropped out (one patient each due to pCLE device failure, food stasis, patient refusal, neuroendocrine tumor, and multiple gastric lesions); therefore, 23 patients were ultimately included. Figure 1 shows the study design and flow.

Written informed consent was obtained from all patients. The study protocol was approved by the Institutional Review Board and the Hospital Research Ethics Committee of Severance Hospital (IRB no. 4-2017-0770).

### 2.2. Study Protocol and Patient Grouping

All endoscopic examinations and biopsies were performed by an endoscopist (S.K.L.) who is an expert in advanced imaging methods, has extensive experience, and has performed more than 100 pCLE procedures.

WLE/M-NBI endoscopy (GIF-HQ290; Olympus Co., Seoul, Korea) was performed as the primary examination of the gastric lesions. After careful observation of the gastric lesions under WLE/M-NBI, sodium fluorescein (2.5 mL; 10%) was injected intravenously. A confocal miniprobe (field of view, 240 μm; lateral resolution, 1 μm; Cellvizio GastroFlex UHD; Mauna Kea Technologies, Paris, France) was then passed through the endoscope accessory channel and placed in gentle contact with the lesion. The Miami classification for interpreting pCLE findings was used to differentiate GC from normal tissues [11]. The existence of residual cancer was defined as the presence of findings such as a completely disorganized epithelium, fluorescein leakage, and a dark irregular epithelium (Figure 2c).

In the WLE/M-NBI examination, a gastric lesion was defined as having residual GC tissue if there was a tumorous mass or an active or healing ulcer with malignant features. We supplementally used the MSVS criteria proposed by Yao et al. and Ezoe et al. for the diagnosis of residual GC tissue with M-NBI [14]. Lesions showing an abnormal vascular network or an abnormal surface pattern with clear demarcation lines were suspected of having residual GC tissue. The suspected area with residual GC tissue observed on WLE/NBI-M was marked for subsequent biopsy, and the lesions were observed using pCLE. Typical cancer cells or abnormal epithelial glands were identified as residual cancers with pCLE and marked for endoscopic biopsy. Most recent studies investigating the optimal number of endoscopic biopsy specimens in lesions suspected as GC suggest that at least three to five biopsies are needed [15]. Therefore, five tissue samples were collected from the gastric lesions with residual GC tissue on both WLE/NBI-M and pCLE. If two marked lesions overlapped, samples were initially collected from those marked on WLE/NBI-M. For lesions in which residual GC was not detected by WLE/NBI-M or pCLE, five tissue samples were randomly collected.

### 2.3. Histopathological Assessment

All deidentified specimens were sent to the pathology department and evaluated by expert gastrointestinal pathologists. The final pathological diagnosis was GC with differentiated or undifferentiated histology, according to the Japanese classification [16]. Undifferentiated types included poorly differentiated adenocarcinoma, signet ring cell carcinoma, and mucinous adenocarcinoma, whereas differentiated types included papillary adenocarcinoma and moderately to well-differentiated adenocarcinoma.

### 2.4. Sample Size Calculation

Although there have been few studies on endoscopic biopsies for diagnosing residual GC, we hypothesized that the sensitivity and specificity of WLE/M-NBI versus pCLE for detecting residual GC lesions would be 65% and 70% and 50% and 80%, respectively. The significance level of α was set at 0.05, and the allowable error of δ was set at 0.1, using a power of 80%. According to the formula for determining sample size, 55 patients were required. We were, however, unable to recruit as many patients as desired.

### 2.5. Statistical Analysis

All statistical analyses were performed using SPSS version 24 (IBM Corp., Armonk, NY, USA). The baseline variables are presented as frequencies and percentages, while descriptive statistics for continuous variables are presented as means and standard deviations. Categorical variables are presented as numbers and proportions.

Sensitivity, specificity, and accuracy were computed for each method along with exact binomial 95% confidence intervals, with histological diagnosis serving as the “gold standard”. Accuracy was defined as follows: (number of true positives + number of true negatives)/(number of true positives + number of true negatives + number of false positives + number of false negatives). McNemar’s test was used to compare differences in sensitivity, specificity, and accuracy between the modalities. Values of *p* < 0.05 were considered significant.

## 3. Results

### 3.1. Patient Characteristics

Baseline patient and lesion characteristics are shown in Table 1. The median patient age was 57.2 ± 10.1 years. In total, 15 men (65.2%) and 8 women (34.8%) were included. A total of 21 patients (91.3%) underwent endoscopy for a response evaluation after palliative chemotherapy, and two patients (8.7%) underwent endoscopy for pre-evaluation to register for the clinical study. All participating patients had tumor-node-metastasis stage IV GC. At the time of the endoscopy, during palliative chemotherapy, complete response according to the modified Response Evaluation Criteria in Solid Tumors was not achieved in any patient, whereas partial response, stable disease, and progressive disease were observed in 4 (17.4%), 15 (65.2%), and 4 (17.4%) patients, respectively.

### 3.2. Accuracy of WLE/M-NBI versus pCLE 

Overall, 80 biopsy samples (57.1%) from 14 patients with residual cancer were histologically diagnosed as adenocarcinoma and dysplasia after WLE/M-NBI/pCLE. The remaining 60 samples (42.9%) were histologically confirmed as non-neoplastic (chronic superficial gastritis, *Helicobacter pylori* gastritis, and atypical cells). We analyzed the accuracy of WLE/M-NBI versus pCLE by collecting all cases for which histological analysis was possible. As shown in Table 2, pCLE showed a significantly higher sensitivity (67% vs. 47%) and negative predictive value (66% vs. 56%) than WLE/M-NBI. Moreover, the accuracy of pCLE was superior to that of WLE/M-NBI (80% vs. 68%, respectively).

Next, we compared the accuracies of WLE/M-NBI and pCLE for residual GC when the results were the same versus when they were contradictory (Table 3).

In patients with identical WLE/M-NBI and pCLE residual GC findings, there was no difference in their predictive ability. However, among patients whose WLE/M-NBI and pCLE results were contradictory, pCLE showed a better predictive ability for residual GC than WLE/M- NBI (66.6% vs. 33.3%). Here, we present a case with Borrmann type IV advanced GC at the time of diagnosis (a). After chemotherapy, no residual cancer was observed with WLE/M-NBI (b). With pCLE, however, a disorganized and irregular thickened epithelium lesion was observed and predicted to have residual GC (c). Five pieces of tissue showed signet ring cell carcinoma (Figure 2).

### 3.3. Comparison of WLE/M-NBI and pCLE by Number of Biopsy Specimens Histologically Confirmed as Containing GC

Eleven lesions were predicted to have residual GC on both WLE/M-NBI and pCLE. In the actual biopsy, the presence of residual GC was proven in 10 of 11 lesions via WLE/M-NBI-targeted biopsy (10/11 [90.9%]). The pCLE-targeted biopsy showed the same results (10/11 [90.9%]). We compared the number of histologically confirmed cancer fragments in the targeted biopsy specimens on WLE/M-NBI versus pCLE (Figure 3). A total of 52.2% (32/55) and 74.5% (41/55) of the WLE/M-NBI- and pCLE-targeted biopsy specimens contained GC fragments, respectively, suggesting that pCLE-targeted biopsy is more likely to yield GC tissue. A mean 3.75 (range, 0–5) pieces contained cancer tissue in the pCLE group—significantly higher than the mean 2.90 (range, 0–5) in the WLE/NBI-M group.

Three lesions were predicted to have residual GC on WLE/M-NBI but not pCLE. On the actual biopsy, residual GC was found in only one of the three lesions. Three lesions were suspected to have residual GC on pCLE but not on WLE/M-NBI. In the actual biopsy, residual GC was found in two of the three lesions. The remaining six lesions were predicted to have no residual GC on either WLE/M-NBI or pCLE. In the actual biopsy, none of the six lesions showed residual GC.

## 4. Discussion

Although the cost and time required to apply the new technology of pCLE require consideration, its effect on the differential diagnosis of malignant and benign gastrointestinal diseases is considerable. There are many areas in which pCLE technology can be helpful in addition to the differential diagnosis of malignant and benign tumors. We have already demonstrated that pCLE-targeted endoscopic biopsy is a suitable molecular biology-based treatment because a greater quantity of GC tissue can be obtained [5,17,18]. Here, we recognized the utility of pCLE in the diagnosis and collection of GC tissues from patients receiving chemotherapy, which is an unmet need in the treatment of patients with advanced GC. In this study, the accuracy of pCLE was superior to that of WLE/M-NBI. NBI has been widely used to detect the focused precancerous condition of GC using high-resolution and wide-field endoscopic images. It is distinguished primarily by its capacity to improve the visibility of superficial mucosa, including microvascular patterns and microsurface structures [6,7]. Furthermore, NBI has demonstrated improved diagnostic usefulness in the detection of precancerous and early neoplastic lesions that might be enhanced in combination with magnification endoscopy [19]. Adding pCLE to WLE/M-NBI significantly improved its predictive ability for residual GC in patients with inconsistent WLE/M-NBI and pCLE results. In this study, pCLE-targeted biopsy also made it possible to obtain more GC tissue, confirming our previous findings.

Post-treatment endoscopic biopsy is a cornerstone for determining the presence of pathological complete remission. It has gradually become known that post-treatment endoscopic biopsy may not help assess the pathological response to treatment [20]. Among the limitations of post-treatment endoscopic biopsy in GC patients, the most significant is that the diagnostic yield of residual cancer is very low regardless of the actual presence of residual cancer. In previous reports, approximately 40% of patients without residual cancer on endoscopic biopsy after concurrent radiation chemotherapy show residual cancer in surgically resected specimens [21,22]. In a study that evaluated the response rate of endoscopy alone without biopsy in 100 patients receiving preoperative chemotherapy, the sensitivity was 65% and the specificity was 50% [23]. 

In almost all cases, gross lesions persisted after chemotherapy based on endoscopic findings; however, the number of cases in which residual cancer tissue was suspected using WLE/M-NBI was relatively low (60.9% [14/23]) in this study. On WLE/M-NBI, residual GC was defined as the presence of a tumorous mass, an active or healing ulcer with malignant features, neovascularization, or an abnormal structural or vascular pattern. Although there are insufficient data regarding the use of these findings to indicate residual GC, the sensitivity and specificity in this study were 86% and 78%, respectively. This result suggests that detecting the presence of residual GC using WLE/M-NBI alone is challenging. Therefore, other techniques are needed to diagnose and obtain samples of residual GC in GC patients receiving chemotherapy.

While many studies have compared the effects of pCLE to virtual chromoendoscopy, such as M-NBI and FICE in esophageal cancer and colon cancer, there have been relatively few investigations on GC. Bok et al. compared the accuracy of pCLE and histological diagnosis for early GC cases eligible for endoscopic submucosal resection. In that study, the accuracy of pCLE (90.7%) was comparable to that of histological diagnosis (85.2%) [24]. In a study of 238 patients, virtual chromoendoscopy-guided pCLE with targeted biopsy significantly increased the diagnostic yield of premalignant or malignant gastric lesions compared to virtual chromoendoscopy with standard biopsy (31.5% vs. 75.1%) [25]. Overall, pCLE appears to be reliable for the diagnosis of gastric lesions. In a meta-analysis of 23 studies evaluating the diagnostic value of CLE for GC lesions, the pooled sensitivity, specificity, and area under the curve were 92% (90–94%), 97% (96–98%), and 0.9774, respectively [26]. So far, no study has compared the effects of pCLE and virtual chromoendoscopy in GC patients following chemotherapy. As mentioned earlier, post-treatment endoscopic biopsy is challenging because surface GC tissues are replaced by ulcerated and hyperplastic tissues in response to prior chemotherapy. Our study has a strength in that it included only GC patients who received chemotherapy; therefore, the diagnostic accuracy of WLE/M-NBI and pCLE for residual cancer was lower than that reported previously. However, the accuracy of pCLE was superior to that of WLE/M-NBI (80% vs. 68%).

Although this is the first study to evaluate the benefits of pCLE for diagnosis and tissue sampling of residual GC in patients receiving chemotherapy, it has several limitations. First, it included a small sample size. We were unable to complete the study with the planned number of patients. In particular, cases with contradictory results between pCLE and WLE/M-NBI are considered suitable for observing the effects of pCLE; however, such cases accounted for only 26% of cases in our study. Second, the generalizability of pCLE may be low. Only a few medical institutions are equipped with pCLE, as its cost-effectiveness has not been verified. Further studies are required to validate these results. Profiling of genetic characteristics through endoscopic biopsy must be performed before chemotherapy—the clinical usefulness of which must still be established.

## 5. Conclusions

Our study showed that pCLE is superior to WLE/M-NBI in terms of accuracy for diagnosing residual cancer in GC patients receiving chemotherapy. In terms of the presence of cancer tissue in the biopsy specimens, pCLE-targeted biopsy was superior to WLE/M-NBI-targeted biopsy. Our results suggest that pCLE should be considered when it is necessary to confirm the presence of residual cancer and obtain tissue samples from patients with GC receiving chemotherapy.

## Figures and Tables

**Figure 1 cancers-14-04319-f001:**
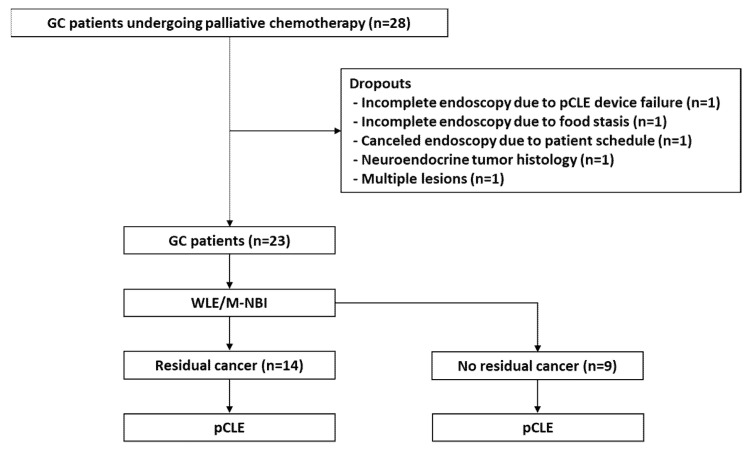
Brief illustration of the study protocol and patient grouping. GC, gastric cancer; M-NBI, magnified narrow-band imaging; pCLE, probe-based confocal laser endomicroscopy; WLE, white-light endoscopy.

**Figure 2 cancers-14-04319-f002:**
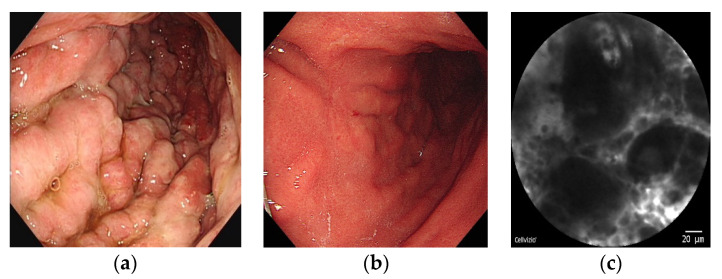
Images of a representative case of residual gastric cancer detected on pCLE. (**a**) Borrmann type IV advanced gastric cancer at initial diagnosis; (**b**) Scarring and subtle thickening of the mucosa after chemotherapy; (**c**) A disorganized and irregular thickened epithelium on pCLE. pCLE, probe-based confocal laser endomicroscopy.

**Figure 3 cancers-14-04319-f003:**
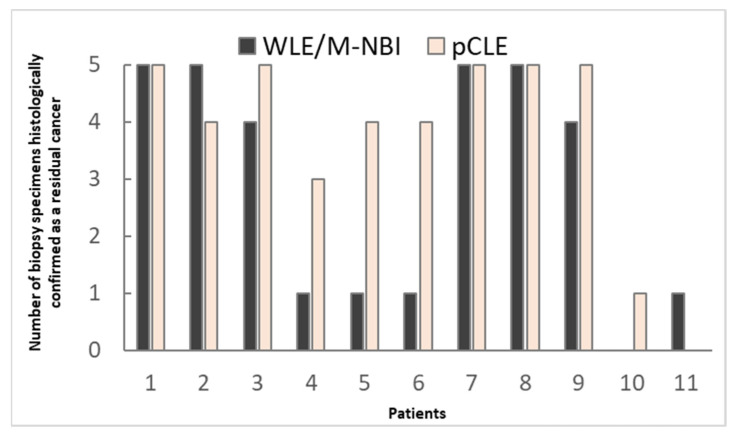
Comparison of the number of biopsy specimens histologically confirmed as containing residual cancer on WLE/M-NBI versus pCLE. GC, gastric cancer; M-NBI, magnified narrow-band imaging; pCLE, probe-based confocal laser endomicroscopy; WLE, white-light endoscopy.

**Table 1 cancers-14-04319-t001:** Characteristics of the enrolled patients (n = 23).

Age (years), mean ± SD	57.2 ± 10.1
Sex, n (%)	
Male	15 (65.2)
Female	8 (34.8)
Reason for endoscopic biopsy, n (%)	
Response evaluation during chemotherapy	21 (91.3)
Registration for clinical study	2 (8.7)
TNM stage, n (%)	
I	0 (0)
II	0 (0)
III	0 (0)
IV	23 (100)
Response after chemotherapy, n (%)	
Complete response	0 (0)
Partial response	4 (17.4)
Stable disease	15 (65.2)
Progressive disease	4 (17.4)

SD, standard deviation; TNM, tumor-node-metastasis.

**Table 2 cancers-14-04319-t002:** WLE/M-NBI versus pCLE for diagnosing residual gastric cancer.

	WLE/M-NBI		pCLE	
	%	95% CI	%	95% CI
Sensitivity	47	0.35–0.59	67	0.55–0.78
Specificity	100	0.92–1.0	100	0.92–1.0
PPV	100	0.89–1.0	100	0.92–1.0
NPV	55	0.43–0.66	66	0.54–0.77
Accuracy	68		80	

CI, confidence interval; NPV, negative predictive value; pCLE, probe-based confocal endomicroscopy; PPV, positive predictive value; WLE/M-NBI, white-light endoscopy/magnifying narrow-band imaging.

**Table 3 cancers-14-04319-t003:** WLE/M-NBI versus pCLE for the diagnosis of residual gastric cancer when the results were the same versus when they were contradictory.

	Predictive Ability of WLE/M-NBI	Predictive Ability of pCLE
Same results	16/17(94.1%)	16/17 (94.1%)
Contradictory results	2/6 (33.3%)	4/6 (66.6%)

pCLE, probe-based confocal endomicroscopy; WLE/M-NBI, white-light endoscopy/magnifying narrow-band imaging.

## Data Availability

The data supporting the findings of this study are available from the corresponding author upon reasonable request.

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
