# Peer review of "Probe-Based Confocal Laser Endomicroscopy versus White-Light Endoscopy with Narrow-Band Imaging for Predicting and Collecting Residual Cancer Tissue in Patients with Gastric Cancer Receiving Chemotherapy"

_cancers, 2022, doi:10.3390/cancers14174319_

Round 1

Reviewer 1 Report

Thank you for allowing me to review this very interesting and important paper. This subject will have increasing importance as the systemic treatment options for gastric cancer continue to improve. We will need to ensure that we can assess response to treatment to be able to recommend surgery or surveillance like we are doing in other cancers that have surgical resections associated with high morbidity (e.g., rectal cancer). I have a few comments to improve the quality of the manuscript:

Page 1 simple summary – acronyms are not defined the first time they are used

Page 1 abstract line 27 – can you include a few words on why chemotherapy makes it challenging and why pCLE would overcome this

Page 1 abstract line 31 – can you include a few words on which patients were selected to undergo both these procedures

Page 2 introduction line 58 – can you provide some of the statistics around the diagnostic yield (sensitivity, specificity etc)

Page 2 methods line 84 – 87  – can you please comment on how the sample size was determined and if consecutive patients were enrolled

Page 3 methods line 115-116 – can you please elaborate on the scientific evidence or rationale for this definition of residual gastric cancer tissue

Page 3 methods line 120-121 – can you please elaborate on the scientific evidence or rationale for this number of biopsies

Author Response

Response to Reviewer 1 Comments

We were most delighted to learn that our manuscript cancers-1848325, entitled “Probe-based confocal laser endomicroscopy versus white-light endoscopy with narrow-band imaging for predicting and collecting residual cancer tissue in patients with gastric cancer receiving chemotherapy”, has been given the opportunity to be resubmitted for publication in cancers. We have carefully considered the valuable comments and suggestions provided by the reviewers and the editor, and made great efforts to improve the manuscript accordingly. The following are our point-by-point answers to specific questions raised by the reviewer. We hope that the revised version of manuscript is now deemed suitable for publication.

  1. Page 1 simple summary – acronyms are not defined the first time they are used

Response) Thank you for your comments. The initial acronyms were defined.

“Collecting appropriate gastric cancer (GC) tissues is critical for molecular biology research or the development of new target drugs for cases resistant to chemotherapy. Chemotherapy may reduce or alter the distribution of GC tissue on the surface, making detection of GC tissue during upper endoscopy challenging. Our study showed that probe-based confocal laser endomicroscopy (pCLE) is superior to white-light endoscopy (WLE) with magnifying narrow-band imaging (M-NBI) in terms of accuracy for diagnosing residual cancer in GC patients receiving chemotherapy. pCLE might be considered when it is necessary to confirm the presence of residual cancer and get tissue samples from GC patients receiving chemotherapy.”

  1. Page 1 abstract line 27 – can you include a few words on why chemotherapy makes it challenging and why pCLE would overcome this

Response) We added a few words what you pointed out in the abstract.

“Chemotherapy, however, may reduce or alter the distribution of GC tissue on the surface, making detection of GC tissue during upper endoscopy challenging. Probe-based confocal laser endomicroscopy (pCLE) is a new technology that enables histological diagnosis by magnifying the mucous membrane to a microscopic level. Here, we evaluated whether pCLE could increase the yield of endoscopic biopsy for GC compared to white-light endoscopy (WLE) with magnifying narrow-band imaging (M-NBI) in GC patients receiving chemotherapy with its powerful imaging technique.”

  1. Page 1 abstract line 31 – can you include a few words on which patients were selected to undergo both these procedures

Response) We added a few words according to your advice in the abstract.

“Patients underwent WLE/M-NBI and pCLE for the detection of residual GC for the purpose of response evaluation or clinical trial registration.”

  1. Page 2 introduction line 58 – can you provide some of the statistics around the diagnostic yield (sensitivity, specificity etc)

Response) We included a sentence about the diagnostic yield statistics according to your recommendation.

“WLE/M-NBI had been reported to have a sensitivity of 93% to 97% for distinguishing GC from benign lesions, while the specificity was 84% to 97% (6, 7).”

  1. Page 2 methods line 84 – 87 – can you please comment on how the sample size was determined and if consecutive patients were enrolled

Response) Although there have been few studies of endoscopic biopsy diagnosing residual GC, we hypothesized that the sensitivity and specificity of WLE/M-NBI versus pCLE for detecting residual GC lesions were 65% and 70%, 50% and 80%, respectively. The significance level of α was set at 0.05, and the allowable error of δ was set at 0.1, using the power of 80%. According to the formula for determining sample size, 55 patients were required. We were, however, unable to recruit as many patients as desired. We included a new paragraph of sample size calculation to the method section.

  1. Page 3 methods line 115-116 – can you please elaborate on the scientific evidence or rationale for this definition of residual gastric cancer tissue

Response) In the WLE/M-NBI examination, a gastric lesion was defined as having residual GC tissue if there was a tumorous mass, an active or healing ulcer with malignant features. We supplementally used the MSVS criteria proposed by Yao et al. and Ezoe et al. for the diagnosis of residual GC tissue with M-NBI. Lesions showing an abnormal vascular network or an abnormal surface pattern with clear demarcation lines were suspected of residual GC tissue. We added a supplementary explanation to the methods section.

  1. Page 3 methods line 120-121 – can you please elaborate on the scientific evidence or rationale for this number of biopsies

Response) Most recent studies to investigate optimal number of endoscopic biopsy specimens in lesions suspected as GC suggested at least three to five biopsies were needed. We added a supplementary explanation to the methods section.

Reviewer 2 Report

This article studied the detection and collection of gastric cancer (GC) tissue in patients receiving chemotherapy with two types of opt-electrical equipment, probe-based confocal laser endomicroscopy (pCLE) and white-light endoscopy with magnifying narrow-band imaging (WLE/M-NBI), while pCLE showed significant advantages. This article contains certain innovations, and the methodology is sound and valid; however, some improvements to the article would be needed:

1.      This article cited lots of previous work on pCLE to describe its advantages. However, a limited number of studies on GC tissues with WLE/M-NBI were mentioned. As a comparison, previous studies especially their detection and diagnosis of GC tissues with WLE/M-NBI should also be discussed in the Discussion section if not the Introduction section.

2.      There are many other researchers who compared pCLE directly to virtual chromoendoscopy including M-NBI, although may not necessarily focus on the detection and diagnosis of gastric cancer. The authors should mention their findings and specify the novelty of this particle compared to previous works.

3.      No need for Simple Summary section, given that it’s the same as Abstract.

4.      Some clarify questions regarding the data present in Table 3. What do the authors define “same results” and “contradictory results” here? Is the accuracy the same as defined in lines 139-141? Looks like the patients are divided into two groups with 17 and 6 of them in each. What’s the reasoning behind this?

5.      Some sentences needed to reword for easy understanding:

·       Long sentences that are difficult to read or misleading. Lines 35-38. The reviewer suggests rewording as:

Moreover, pCLE showed better predictive ability for residual GC than WLE/M-NBI, when WLE/M-NBI and pCLE showed inconsistent results. pCLE diagnosed residual GC more accurately than WLE/M-NBI, which resulted in an increased number of GC tissues collected during the endoscopic biopsy.

·       In a lot of long sentences, “patients with GC” can be simplified as “GC patients”.

·       Line 172: “Among patients in whom WLE/M-NBI and pCLE showed the same results for residual”. Same for line 173: “….. However, among patients …”.

6.      Incorrectly labeled sections:

·       Line 132 should be “2.4. Statistical analysis”

·       Line 262 should be “5. Conclusions”

7.      Are all the figures in Figure 2 from pCLE or some from pCLE some from WLE/M-NBI? It’s confusing together with the description in lines 175-178.  

Author Response

Response to Reviewer 2 Comments

We were most delighted to learn that our manuscript cancers-1848325, entitled “Probe-based confocal laser endomicroscopy versus white-light endoscopy with narrow-band imaging for predicting and collecting residual cancer tissue in patients with gastric cancer receiving chemotherapy”, has been given the opportunity to be resubmitted for publication in cancers. We have carefully considered the valuable comments and suggestions provided by the reviewers and the editor, and made great efforts to improve the manuscript accordingly. The following are our point-by-point answers to specific questions raised by the reviewer. We hope that the revised version of manuscript is now deemed suitable for publication.

  1. This article cited lots of previous work on pCLE to describe its advantages. However, a limited number of studies on GC tissues with WLE/M-NBI were mentioned. As a comparison, previous studies especially their detection and diagnosis of GC tissues with WLE/M-NBI should also be discussed in the Discussion section if not the Introduction section.

Response) Thank you for your insightful comments. In the discussion section, we included earlier research about the detection and diagnosis of GC with WLE/M-NBI.

“NBI has been widely used to detect the focused precancerous condition of GC using high-resolution and wide-field endoscopic images. It is distinguished primarily by its capacity to improve visibility of superficial mucosa, including microvascular patterns and microsurface structures (6, 7). Furthermore, NBI has demonstrated improved diagnostic usefulness in the detection of precancerous and early neoplastic lesions that might be enhanced in combination with magnification endoscopy (19).”

  1. There are many other researchers who compared pCLE directly to virtual chromoendoscopy including M-NBI, although may not necessarily focus on the detection and diagnosis of gastric cancer. The authors should mention their findings and specify the novelty of this particle compared to previous works.

Response) According to your recommendation, we emphasized the strength of our study once more in the discussion section.

While many studies have compared the effects of pCLE to virtual chromoendoscopy such as M-NBI and FICE in esophageal cancer and colon cancer, there have been relatively few investigations on gastric cancer. In a recent study of 238 patients, virtual chromoendoscopy-guided pCLE with targeted biopsy significantly increased the diagnostic yield for the detection of premalignant or malignant gastric lesions compared to virtual chromoendoscopy with standard biopsy (31.5% vs. 75.1%) (25). So far, no study has compared the effects of pCLE and virtual chrmoendoscopy in GC patients following chemotherapy. As mentioned earlier, post-treatment endoscopic biopsy is challenging because surface GC tissues are replaced by ulcerated and hyperplastic tissues in response to prior chemotherapy. Our study has a strength in that it included only GC patients who received chemotherapy; therefore, the diagnostic accuracy of WLE/M-NBI and pCLE for residual cancer was lower than that reported previously. However, the accuracy of pCLE was superior to that of WLE/M-NBI (80% vs. 68%).”

  1. No need for Simple Summary section, given that it’s the same as Abstract.

Response) Thank you for your keen comments. We modified simple summary according to your comments.

“Collecting appropriate gastric cancer (GC) tissues is critical for molecular biology research or the development of new target drugs for cases resistant to chemotherapy. Chemotherapy may reduce or alter the distribution of GC tissue on the surface, making detection of GC tissue during upper endoscopy challenging. Our study showed that probe-based confocal laser endomicroscopy (pCLE) is superior to white-light endoscopy (WLE) with magnifying narrow-band imaging (M-NBI) in terms of accuracy for diagnosing residual cancer in GC patients receiving chemotherapy. pCLE might be considered when it is necessary to confirm the presence of residual cancer and get tissue samples from GC patients receiving chemotherapy.”

  1. Some clarify questions regarding the data present in Table 3. What do the authors define “same results” and “contradictory results” here? Is the accuracy the same as defined in lines 139-141? Looks like the patients are divided into two groups with 17 and 6 of them in each. What’s the reasoning behind this?

Response) When WLE/M-NBI and pCLE made the same diagnosis for residual GC or not, we defined it the same result (n=17). When they made the opposite diagnosis, we defined it the contradictory result (n=6). When WLE/M-NBI and pCLE showed contradictory results, pCLE showed better predictive ability for residual GC than WLE/M-NBI (66.6% vs. 33.3%).  As you pointed out, ‘accuracy’ in table 3 does not meet the definition we defined. Therefore, we modified a few words in table 3.

Table 3. WLE/M-NBI versus pCLE for the diagnosis of residual gastric cancer when the results were the same versus contradictory

Predictive ability of WLE/M-NBI

Predictive ability of pCLE

Same results

16/17(94.1%)

16/17 (94.1%)

Contradictory results

2/6 (33.3%)

4/6 (66.6%)

pCLE, probe-based confocal endomicroscopy; WLE/M-NBI, white-light endoscopy/magnifying narrow-band imaging

  1. Some sentences needed to reword for easy understanding:
  • Long sentences that are difficult to read or misleading. Lines 35-38. The reviewer suggests rewording as:

Moreover, pCLE showed better predictive ability for residual GC than WLE/M-NBI, when WLE/M-NBI and pCLE showed inconsistent results. pCLE diagnosed residual GC more accurately than WLE/M-NBI, which resulted in an increased number of GC tissues collected during the endoscopic biopsy.

  • In a lot of long sentences, “patients with GC” can be simplified as “GC patients”.
  • Line 172: “Among patients in whom WLE/M-NBI and pCLE showed the same results for residual”. Same for line 173: “….. However, among patients …”.

Response) Thank you for your kind comments. We modified what you suggested.

“Moreover, pCLE showed better predictive ability for residual GC than WLE/M-NBI, when WLE/M-NBI and pCLE showed inconsistent results. pCLE diagnosed residual GC more accurately than WLE/M-NBI, which resulted in an increased number of GC tissues collected during the endoscopic biopsy.”

In patients with identical WLE/M-NBI and pCLE residual GC findings, there was no difference in their predictive ability. However, among patients whose WLE/M-NBI and pCLE results were contradictory, pCLE showed a better predictive ability for residual GC than WLE/M- NBI (66.6% vs. 33.3%).

  1. Incorrectly labeled sections:
  • Line 132 should be “2.4. Statistical analysis”
  • Line 262 should be “5. Conclusions”

Response) We corrected what you pointed out.

  1. Are all the figures in Figure 2 from pCLE or some from pCLE some from WLE/M-NBI? It’s confusing together with the description in lines 175-178.

Response) We modified what you pointed out in line 175-178.

“Here, we present a case with Borrmann type IV advanced GC at the time of diagnosis (a). After chemotherapy, no residual cancer was observed with WLE/M-NBI (b). With pCLE, a disorganized and irregular thickened epithelium lesion was observed and predicted to have residual GC (c). Five pieces of tissue showed signet ring cell carcinoma (Figure 2).”

Reviewer 3 Report

This study compared two modalities (Probe-based confocal laser endomicroscopy versus white-light endoscopy with narrow-band imaging) to identify residual cancer tissue in patients with gastric cancer receiving chemotherapy. I have somes suggestions listed below. 

1. This study was well written and organized. However, I wonder how many cases should be collected to draw the conclusion interms of study power.

2. Although the better detection rate of redisual cancer was noted by the pCLE, it might not change the clinical implication/practive. In the follow-up cancer without residual cancer, I am not sure the gastrectomy will be omitted ornot.   

Author Response

Response to Reviewer 3 Comments

We were most delighted to learn that our manuscript cancers-1848325, entitled “Probe-based confocal laser endomicroscopy versus white-light endoscopy with narrow-band imaging for predicting and collecting residual cancer tissue in patients with gastric cancer receiving chemotherapy”, has been given the opportunity to be resubmitted for publication in cancers. We have carefully considered the valuable comments and suggestions provided by the reviewers and the editor, and made great efforts to improve the manuscript accordingly. The following are our point-by-point answers to specific questions raised by the reviewer. We hope that the revised version of manuscript is now deemed suitable for publication.

  1. This study was well written and organized. However, I wonder how many cases should be collected to draw the conclusion interms of study power.

Response) Thank you for your kind comments. Although there have been few studies of endoscopic biopsy diagnosing residual gastric cancer, we hypothesized that the sensitivity and specificity of ME-NBI versus CLE for detecting residual gastric cancer lesions were 65% and 70%, 50% and 80%, respectively. The significance level of α was set at 0.05, and the allowable error of δ was set at 0.1, using the power of 80%. According to the formula for determining sample size, 55 patients were required. We were, however, unable to recruit as many patients as desired. We included a new paragraph of sample size calculation to Method section.

  1. Although the better detection rate of redisual cancer was noted by the pCLE, it might not change the clinical implication/practive. In the follow-up cancer without residual cancer, I am not sure the gastrectomy will be omitted or not.

Response) The main purpose of comparing pCLE and m-NBI in patients receiving chemotherapy in our study is to compare the accuracy of the diagnosis of residual cancer. As the reviewer mentioned, it is difficult to answer with the results of this study to determine whether gastrectomy is required in patients with no redisual cancer in pCLE.

Round 2

Reviewer 2 Report

The current version looks good, except that some minor adjustments may be needed:

1.       Part 5 “Conclusions” and part 6 “References” need to be reformated.

2.       Original comment 7 regarding Figure 2. According to the adjustments of lines 175-178 in version 1 (sorry, it doesn’t show the line numbers in version 2), Fig.2 b is with WLE/M-NBI, Fig2 c is with pCLE, Fig 2. a is still not clear. The captions for Fig.2 2 should also be edited accordingly.

Reviewer 3 Report

My comments had been addressed comprehensively.